# Physical Education, Quality of Breathable Air and Their Effects on the Formation of High School Students as Sustainability in Maintaining the Lifestyle. Is the Physical Education Lesson Enough to Create Such Valences?

**Camelia Plastoi** [1,*,†], **Ioana Butu** [2,†], **Diana-Mihaela Țîrcă** [3,†], **Bianca Ferrario** [4,†], **Ilie Mihai** [5,†], **Daniel Chivu** [1,†] **and Eduard Leonard Guță** [1,†]

1   Health and Motricity Department, "Constantin Brâncuși" University of Targu-Jiu, 210135 Targu-Jiu, Romania; daniel.chivu1984@gmail.com or chivu.daniel@utgjiu.ro (D.C.); leonard_eduard_85@yahoo.com or guta.leonard@utgjiu.ro (E.L.G.)

2   Physical Education, Sport and Kinesiotherapy Department, Spiru Haret University, 041905 Bucharest, Romania; mi2oana@yahoo.com or ushefs_butu.oana@spituharet.ro

3   Management and Businesss Administration Department, "Constantin Brâncuși" University of Târgu-Jiu, 210135 Targu-Jiu, Romania; diana.mihaela.tirca@gmail.com or diana@utgjiu.ro

4   Physical Education and Sport Department, Valahia University of Targoviste, 130105 Targoviste, Romania; ferrariobianca@yahoo.com

5   Physical Education and Sport Department, University of Pitesti, 110040 Pitesti, Romania; ilie112004@yahoo.com or ilie.mihai@upit.ro

*   Correspondence: cami_plastoi@yahoo.com or camelia.plastoi@utgjiu.ro; Tel.: +40-722720640

†   All authors have equal contributions to this work.

**Abstract:** This study was conducted to emphasize the correlation between the number of physical education lessons with effects on the formation of high school students and the importance of practicing physical activities during the extra-class time in sustainable spaces with quality breathable air. The values were recorded in this cross-sectional study; on a number of 208 high school students; grades 9–12; tested for the level of manifesting their effort capacity acquired during the physical education lessons. The statistical analysis of the processed data highlights the obtained values: Weak 59% and 31% satisfactory for boys; and 53.92% weak; 34% satisfactory for girls. In conclusion; the necessity to carry out extra-curricular physical activities in an unpolluted environment and to acquire useful physical skills in carrying out different activities in society, promotes and generates sustainability in maintaining the lifestyle.

**Keywords:** acquiring new information skills; effort capacity; lifestyle; physical abilities environmentally friendly

---

## 1. Introduction

The environment in which people live is defined firstly by the quality of air, water, soil, dwelling place, and by the food they eat, as well as by the environment where they carry on their habitual activities. Closely related to these factors, influenced and determined in the short or long run by them, is the population's health condition [1]. In their vast majority, green spaces are arranged for definite purposes and are used for certain activities, the laws in force supporting their intentions. "Generally, the city is a great resources consumer and a big producer of pollutant emissions and waste, being therefore a great threat and influence on the environment" [2].

The effects on increasing air pollution can be reduced by involving the population in programs such as "Sport for people", programs that result in body resistance, and increase of the influence of environmental factors. As a guarantee for a good future for the population, it is necessary that future preschool children and pupils should be involved in a lot of sporting activities.

At the same time, it is very important that the times and the places where the physical activities are carried out should be corroborated with the periods of the day when the values of the pollutants are as low as possible [1,2] "various efforts being undertaken in order to achieve a decarbonized, energy secured road transport system" [3].

Education and responsibilities start from an early age and are part of a permanent, lifelong process, "therefore, this is a dynamic process and that associates the product of the interaction of memory with the awareness of certain values" [4,5].

The process by which high school students (HSS) try to form their own views and ideas about all that is happening in life is a delicate, controversial, and oscillating one, relying, most of the time, on the influence exerted by "parents, teachers and age group" [6], therefore, their being endowed with different abilities, achieved by persons trained in this respect/in a professional setting, of carrying out various physical activities in order to maintain a healthy lifestyle, is considered to be very important in this age period. In this regard, there are different programs of "the promotion and support of educators, individually and collectively" [7] under "the action of formal and informal institutions, social capital may provide unique insights about this important process" [8], generating innovations in their teaching and education.

In order to be able to support "the sustainability of the environment in the workplace" [9] in the long run through a correct and realistic development, we must mention that "the success of high school students, in a professional plan, is conditioned by the biological, motor resources of the individual (the motor capacity, the physical condition)" [10], closely related to the personal health acquired in an unpolluted environment at this age and to the time allotted to physical activities in the school curriculum. Within this context, the research conducted in recent years in Romanian education highlights aspects related to the insufficient number of hours of physical education [11,12], which generates the allocation of an additional time, from the free time of the HSS in order to maintain a healthy lifestyle.

Counteracting the negative social effects that HSS are exposed to, as well as creating a balance between daily effort made (intellectual, verbal, physical) can be achieved through correct management of their activity in order to form sustainable and applicable skills throughout life. Thus, physical education and sports, practiced at school age, complying with a pattern of "structure-process-outcome which should identify the resources that influence the frequency of PE and intensity of physical activity during PE" [13], generate sustainable effects in support of different efforts in the medium run (at school and in free time) and in the long run (at work, as a future employee).

"The number of motor skills and general abilities on which an individual is capable to use and put them into practice represent the true measurement of his skills" [14], however, considering that skills are acquired throughout life, their development involves learning, consolidation, and improvement, and the more the student/person is involved in an adequate educational process, the more these skills will help him to be very well prepared. It should be mentioned that, in obtaining high performances (in any field), only genetic qualities are not enough, the multidisciplinary training being the basis for defining and maturing the students/persons. These general aspects also have valid implications in the physical education of the HSS, where the forming of a general physical capacity according to their age, as well as the education of the volition, perseverance, etc., shape their character and the level executions of the elements specific to the different physical, intellectual, and sporting activities, in which "a criterion related to the way the demonstration is designed and compliance the correct execution of the elements regarding the technical and biomechanical aspects" [15] is required throughout the schooling period. Thus, there is a need to develop the training contents, because most training contents make users easily bored due to lack of immersion and fun factors, slowing down repetitive training function [16].

HSS have the ability to make a sustained physical effort, which involves "a mechanical work with the highest intensity, with a duration as long as possible" [17], unconcerned of its forms of manifestation. "The effort's capacity represents the maximum power of which an individual is capable to realize, it being the maximum intensity of the effort that can be realized by a subject, or representing the maximum working quantity of a mechanical thing that can be made in the time unity of an individual" [18].

Physical effort made by high school students is generated by the functionality of "internal organs and systems having at the base the specific energetic potential, from which it results a mechanical thing expressed in acknowledged measurement units" [19], "to which is added a cognitive loading on functional activities" [20], as well as the maintaining of "the body position as natural as possible" [21], which are dependent on the quality of the air breathed in the environment. Thus, the human movement achieved by muscular contraction [22] is the basis of the manifestations of the different practical skills acquired during school hours, especially during the physical education lesson.

Internationally, the importance of the existence of an adequate and balanced curriculum by which the number of hours of physical education for young pupils is materialized in many objectives, like getting good health, good physical development, a healthy lifestyle, and last but not least, pleasure to practice exercises independently or in a team. However, in Romania, most studies emphasize the importance of physical education classes without exemplifying the variety of activities that could be included in the school curriculum, with an important role in acquiring the skills for HSS [23].

The lack of appropriate physical activity, especially during the growth period, exposes the students to different diseases [24,25], which subsequently need to be treated by methods of recovery and rehabilitation [26], and the incidence of air quality, especially in urban areas where, due to varied anthropoid activities, "high concentration of the pollutants in atmosphere are resulting (example: carbon oxides, nitrogen, sulphur, suspension powders, and sedimentary powders, toxic substances) over the limits where they are usually found in the atmosphere" [27,28], generate the reorientation towards spaces or places that offer sustainability, in which various extra-class physical activities can be carried out. These locations are equipped with electronic means that facilitate learning [29] by visualizing personal biomechanical movements, also contributing to the development of positive attitudes towards physical education (PE) [30].

### *1.1. Aim*

The aim of the study was to emphasize the connection between the manifestation of the level of anaerobic effort capacity of high school students during physical education lessons and the quality of breathable air where they practice their sustainable physical activities in order to maintain a healthy lifestyle.

### *1.2. Hypothesis*

The hypothesis from which we started is to emphasize the direct connection between the number of hours of physical education provided in the Romanian school syllabus for high school students with effects on their formation and the necessity to do extra-class physical activities which provide sustainability in maintaining the lifestyle.

## 2. Methodology

The research was conducted at "Mihai Viteazul" National College from Bucharest, during a school year, the subjects being HSS from the 9th–12th grades, in number of 208:105 girls, 103 boys.

The research was carried out by involving the physical education teachers who taught at the study centers and the authors of the paper, who have acquired experience both in teaching physical education in high school and in higher education, their teaching and research experience and research confirming the familiarity of the authors of the paper with the tests used to validate the hypothesis of the study.

The experiment allowed the application of observational and experimental methods, with the purpose of recording and interpreting the values obtained by the HSS, during the physical education lesson (PEL), as well as generating feedbacks of major significance in the management of the didactic process and of the weights from the physical activities in the PEL. This fact allowed us to make a correct quantification of the applied tests [20,31,32].

The recording of the anthropometric data of each student, by age categories were recorded as follows: for girls:

- height (cm) 155 ± 5, weight (kg) 54 ± 5: 14 years–7 girls; 15 years–4 girls; 16 years–3 girls; 17 years–3 girls; 18 years–3 girls, total number 20 girls;
- height (cm) 161 ± 5, weight (kg) 60 ± 5: 14 years–6 girls; 15 years–8 girls; 16 years–7 girls; 17 years–9 girls; 18 years–8 girls, total number 38 girls;
- height (cm) 167 ± 5, weight (kg) 66 ± 5: 14 years–3 girls; 15 years–4 girls; 16 years–8 girls; 17 years–8 girls; 18 years–7 girls, total number 30 girls;
- height (cm) 171 ± 5, weight (kg) 72 ± 8: 14 years–3 girls; 15 years–2 girls; 16 years–4 girls; 17 years–5 girls; 18 years–3 girls, total number 17 girls;

    for boys:

- height (cm) 163 ± 5, weight (kg) 60 ± 5: 14 years–4 boys; 15 years–3 boys; 16 years–3 boys; 17 years–2 boys; 18 years–2 boys, total number 14 boys;
- height (cm) 169 ± 5, weight (kg) 71 ± 5: 14 years–5 boys; 15 years–7 boys; 16 years–8 boys; 17 years–10 boys; 18 years–6 boys, total number 36 boys;
- height (cm) 175 ± 5, weight (kg) 77 ± 5: 14 years–4 boys; 15 years–6 boys; 16 years–8 boys; 17 years–9 boys; 18 years–11 boys, total number 38 boys;
- height (cm) 181 ± 8, weight (kg) 83 ± 8: 14 years–2 boys; 15 years–3 boys; 16 years–4 boys; 17 years–3 boys; 18 years–3 boys, total number 15 boys.

The processing of the data obtained and the use of the Sargent test, (expressed in kg/s) allowed us to evaluate the anaerobic capacity, manifested by the HSS, during the PEL, in accordance with the quality of the air breathed during the physical activities [17]. "Nevertheless, in order to achieve these results, teachers need to design and apply pedagogical strategies routinely, and pay very close attention to the dynamics created in the classroom" [33] and to the responses to visual stimuli (visual reflex), which contribute to the achievement of the tasks appropriate to the formation of psycho-motricity by functioning of the mechanisms of inverse communication (the emission of feed-back) [34].

The high school students' (HSS) motor skills acquisition is at this age at a certain level, and demonstrates personal achievements under the influence of the various factors that have contributed to the development of the student's abilities up to this moment, skills that can be further improved through practical exercises [35]. Therefore, the subjects under study will carry out, without deviating from the structure of the PEL (physical education lesson) the following:

- Recording the anthropometric characteristics of each HSS (kg) (without affecting the time allotted to PEL);
- performing a warm-up of the muscles through specific PEL exercises: Tiptoe with the arms stretched above the head (3 × 15 m, with return to the starting place by relaxing the limbs-active pause); walk on heels with the arms behind the back (3 × 15 m); run with heel toe walking (3 × 20 m); run with the knees up (3 × 20 m); run with heels to the buttocks (3 × 20 m); move by leaping step (3 × 15 m); move with jumping step (3 × 15 m);

- performing 3 series of 3 vertical jump, successive, with a pause of 10 s after each series of vertical jumps. The best vertical jump will be taken into consideration, confirming the level each student has achieved. The best leap will be the result of a correct coordination between the positions of the body segments and breathing, throughout jumping [36].

The obtained values have been related to the afferent standards of age, as follows "boys: Weak grade < 113; satisfactory between 113–149; medium between 150–187; good between 188–224; very good > 224. Girls: Weak grade < 92; satisfactory between 92–120; medium between 121–151; good between 152–182; very good > 182", according to Dal Monte (1998), taken from Niculescu's book [36].

Assessments made and the values obtained from the applied tests were corroborated with the investigation, during the same period, of the values of the concentrations from the different compounds of the breathable air in the locations closest to the place where the study was carried out.

## 3. Statistical Analysis

With the obtained results, we calculated the arithmetic mean (X), the standard deviation (S), coefficient of variation (Cv%), the estimation of medium error (EME), and of the significance of the difference between the two means—the "z" test [35].

The estimation of medium error (EME) represents the margin of error added in the range value 10.83–10.89 (9.84 + 0.99; 9.90 + 0.99) for boys, and for girls, in the range value 11.40–11.51 (10.38 + 1.02; 10.49 + 1.02) (Table 1).

The arithmetic mean (X) for girls has the values: 10.45 (class IX); 9.97 (class X); 10.88 (class XI); 10.25 (class XII), and for boys: 10.66 (class IX); 9.97 (class X); 9.65 (11th grade); 9.09 (class XII), values based on the reference value, separately for girls and boys.

The standard deviation (S) was calculated in the Excel program by using the statistical STDEV function, obtaining values for girls of 93.17 (class IX); 90.44 (class X); 88.21 (11th grade); 87.01 (class XII), and for boys 89.13 (class IX); 95.67 (X class); 112.09 (11th grade); 109.98 (12th grade).

The coefficient of variation (Cv%) calculated by the formula CV = S/X recorded values of 0.08 (class IX); 0.09 (class X); 0.8 (class XI); 0.8 (class XII) for girls, showing a very high homogeneity, and for boys: 0.08 (class IX); 0.09 (class X); 0.11 (class XI); 0.12 (class XII) showing moderate homogeneity.

The "z" test value = 9.02 obtained, for HSS, and compared to the "t" value in Fischer's table, is higher, and taking into account the value threshold $p < 0.01$, we find the difference obtained, thus validating the hypothesis of the study.

The values obtained by appliance of the formula $P = \sqrt{4.95 * G} * \sqrt{D}$, where 4.95 is a constant, G is body weight, D is the detent, expressed as a percentage, for boys are 59% (weak), 31% (satisfactory), and 10% (medium) and for girls are 53.92% (weak), 34% (satisfactory), and 12.08% (medium) (Table 2).

## 4. Results

For a period of one year, the records of these compounds showed values between certain limits, their monitoring being performed also at intervals of 6 months, as well as 6 months after the completion of the tests. The admitted limits recorded at Mihai Bravu point/center, closest to the place where the study was conducted, show the concentrations during a school year of NOx (Figure 1), NO (Figure 2), $SO_2$ (Figure 3), CO (Figure 4) and on 6 month intervals for NOx (Figure 5), CO (Figure 6), and $NO_2$ (Figure 7).

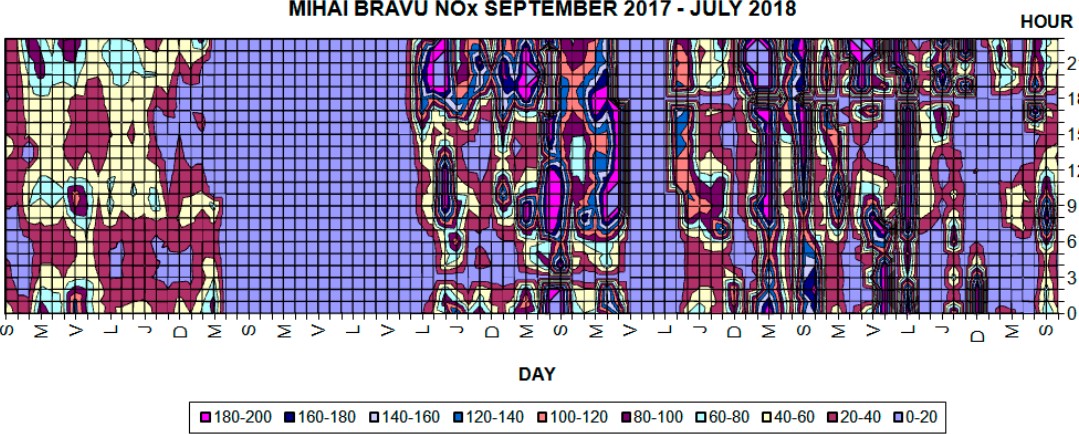

**Figure 1.** The values recorded in the breathable air, during the working hours, regarding the concentration of NOx.

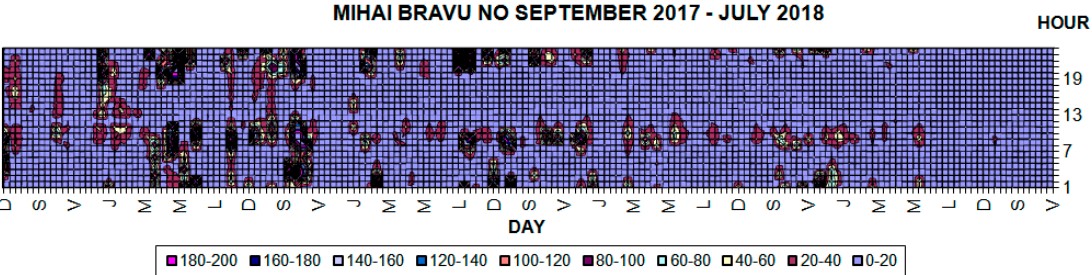

**Figure 2.** The values recorded in the breathable air, between the hours 1 and 19, regarding the concentration of NO.

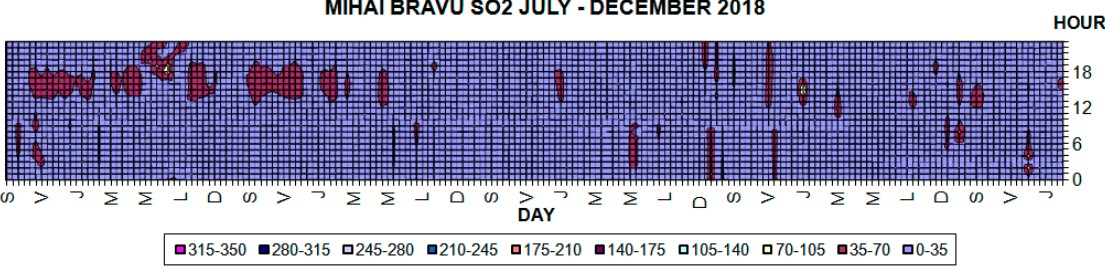

**Figure 3.** The values recorded in the breathable air, between the hours 1 and 18, regarding the concentration of $SO_2$.

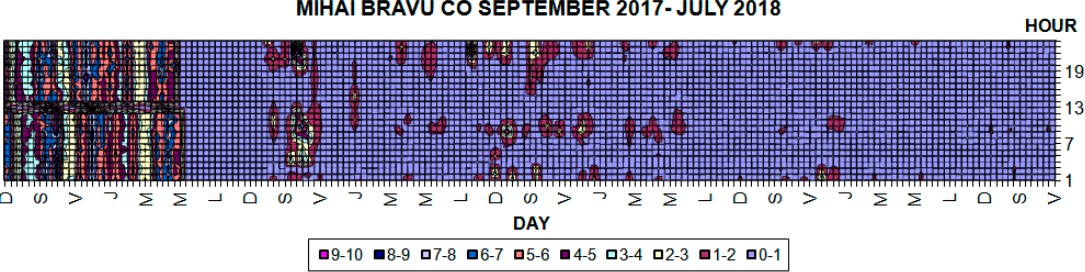

**Figure 4.** The values recorded in the breathable air, during a school year, regarding the concentration of CO.

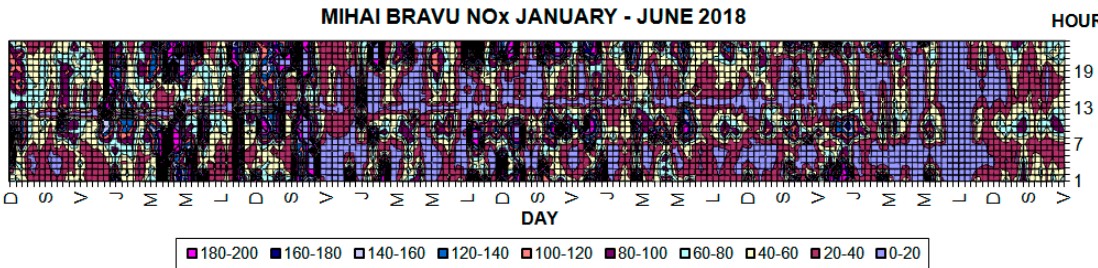

**Figure 5.** The values of NOx recorded for 6 months, between the hours 1 and 19.

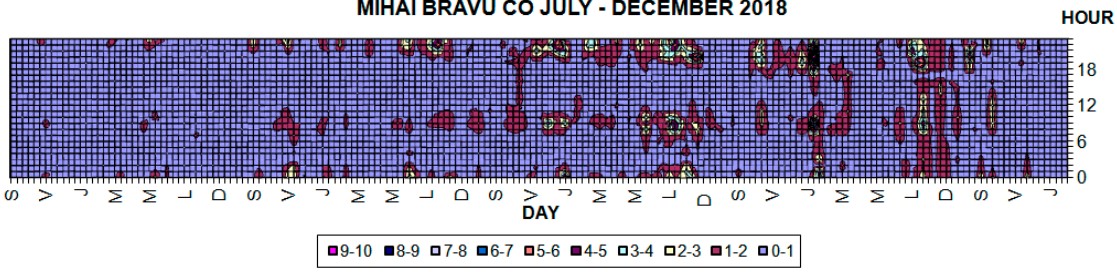

**Figure 6.** The values of CO recorded for 6 months, between the hours 1 and 18.

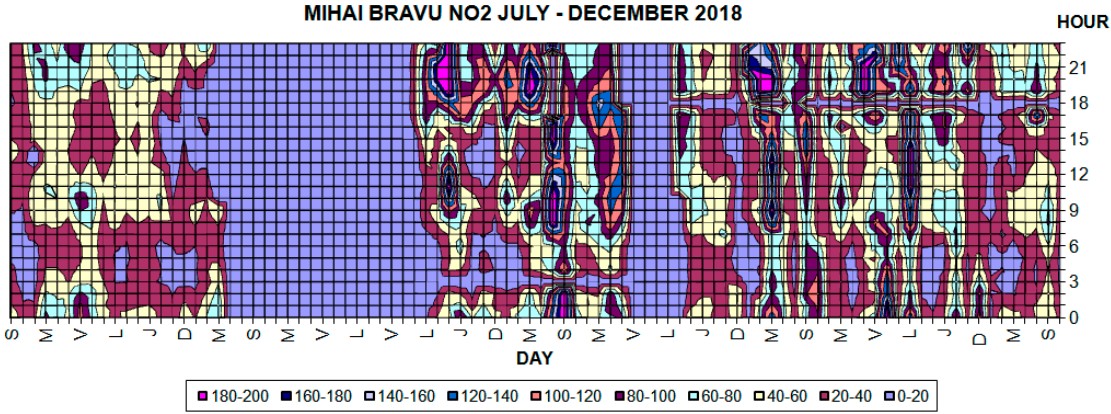

**Figure 7.** The values of $NO_2$ recorded for 6 months, between the hours 1 and 21.

Irrespective of the class level, the values recorded are low (Figure 8), an aspect that highlights the reduced time allotted to physical activities by HSS, insufficient for the acquisition of new skills and a healthy lifestyle.

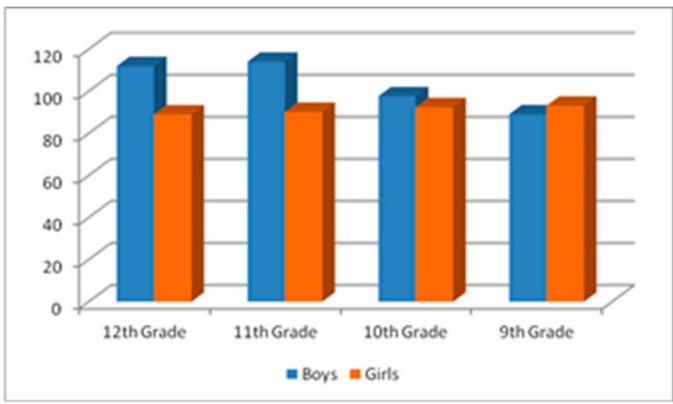

**Figure 8.** The arithmetic values of capacity for anaerobic effort for girls and boys, from IX to XII grades.

The values obtained from the statistical indicators (X, S, Cv, EME, and the "Z" test), recorded in Table 1, and their comparison with the reference values, separately for girls and boys, for *p* < 0.01, show the level of physical effort made by the HSS, related to the time allocated for physical activities. Only the time during the PEL is taken into account and, as the obtained values show, the available time is not enough to develop in HSS an adequate effort capacity to maintain an optimal long-term health condition. The physical activities performed inside of the PEL allowed us to highlight how successful or not this lesson is in acquiring a general capacity of the human body, directly reflected in the lifestyle. The values obtained, expressed as a percentage, are highlighted in Table 2, Figure 9 for boys and in Table 2, Figure 10, for girls.

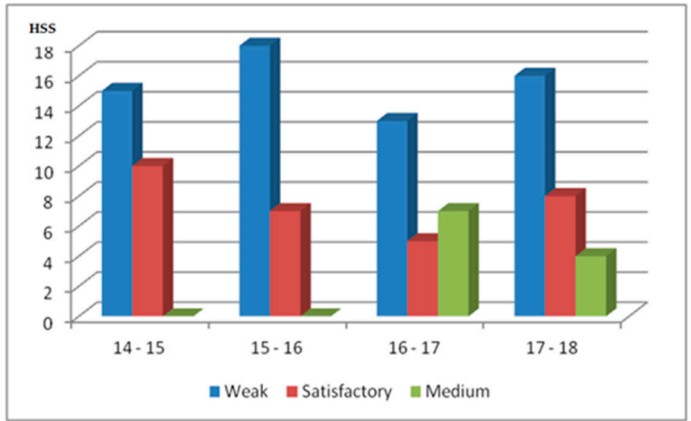

**Figure 9.** Distribution by age groups and sex of young students and their results obtained at the Sargent test, on levels of appreciation (for boys).

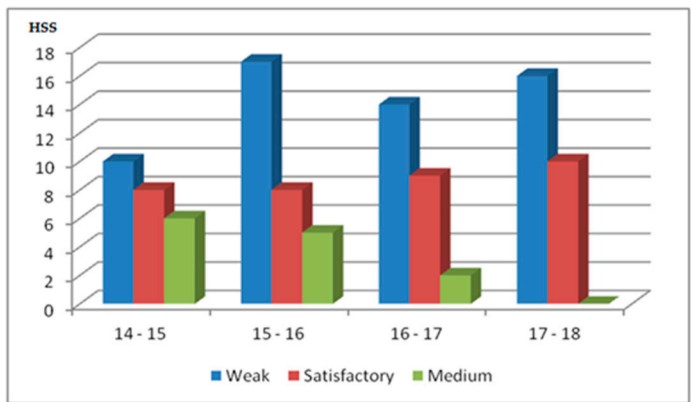

**Figure 10.** Distribution by age groups and sex of young students and their results obtained at the Sargent test, on levels of appreciation (for girls).

**Table 1.** Calculated statistic indicators—the values for boys and girls per class group and sex, reported to the pattern of the statistical indicator for HSS.

| Statistic Indicators | Boys/Grade | | | | | Girls/Grade | | | | |
|---|---|---|---|---|---|---|---|---|---|---|
| | IX | X | XI | XII | Pattern | IX | X | XI | XII | Pattern |
| X | 10.66 | 9.97 | 9.65 | 9.09 | 9.90 | 10.45 | 9.97 | 10.88 | 10.25 | 10.49 |
| S | 89.13 | 95.67 | 112.09 | 109.98 | 101.70 | 93.17 | 90.44 | 88.21 | 87.01 | 89.07 |
| Cv% | 0.08 | 0.09 | 0.11 | 0.12 | 0.18 | 0.08 | 0.09 | 0.08 | 0.08 | 0.12 |
| EME | | | | | 0.99 | | | | | 1.02 |
| "Z" Test | | | | | 9.02 | | | | | |

The values obtained, expressed as a percentage, for boys in Table 2, Figure 9, and for girls in Table 2, Figure 10 demonstrate the insufficient time allotted to the hours of physical education in the school curriculum in Romania.

**Table 2.** Number of subjects and their percentages obtained—appreciation scale for boys and girls per age group (on a scale from weak to medium).

| Level | Age-Boys (% HSS) | | | | | Age-Girls (% HSS) | | | | |
|---|---|---|---|---|---|---|---|---|---|---|
| | 14/15 | 15/16 | 16/17 | 17/18 | 14/18 | 14/15 | 15/16 | 16/17 | 17/18 | 14/18 |
| Weak | 15 | 18 | 13 | 16 | 59% | 10 | 17 | 14 | 16 | 53.92% |
| Satisfactory | 10 | 7 | 5 | 8 | 31% | 8 | 8 | 9 | 10 | 34% |
| Medium | - | - | 7 | 4 | 10% | 6 | 5 | 2 | - | 12.08% |

In order to determine the concentration evolution for nitrogen oxides concentration for 4 data collection centers (Berceni, Drumul Taberei, Lacul Morii, and Mihai Bravu, Figure 11), during one year the data were calculated mathematically. Studies conducted thus to underline a weekly concentration distribution and also an hourly distribution.

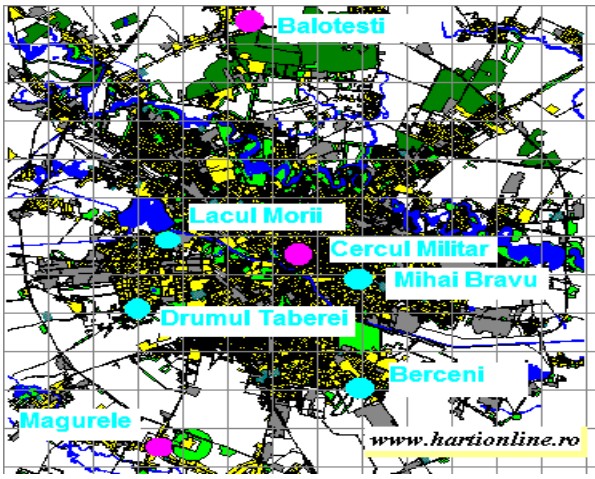

**Figure 11.** Localization for collecting points in Bucharest.

The data were also calculated to underline the cumulative evolution of (CU) pollutants concentration being recorded daily during the study period. In Figures 12–14, the evolution of NO, $NO_2$, $NO_x$ concentration for 4 collecting data centers is comparatively presented.

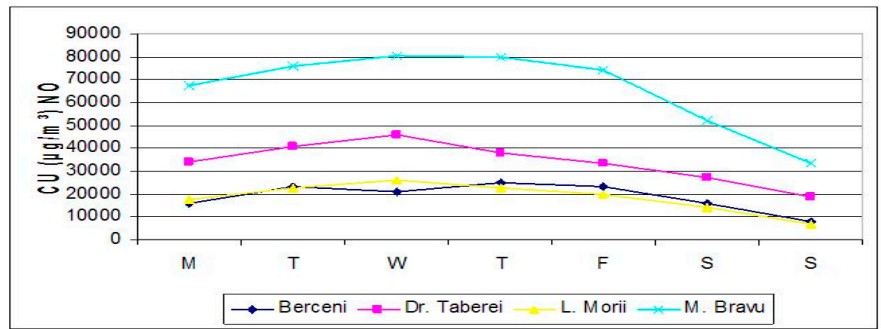

**Figure 12.** Evolution of accumulated quantities in different centers (CU) for NO, in days.

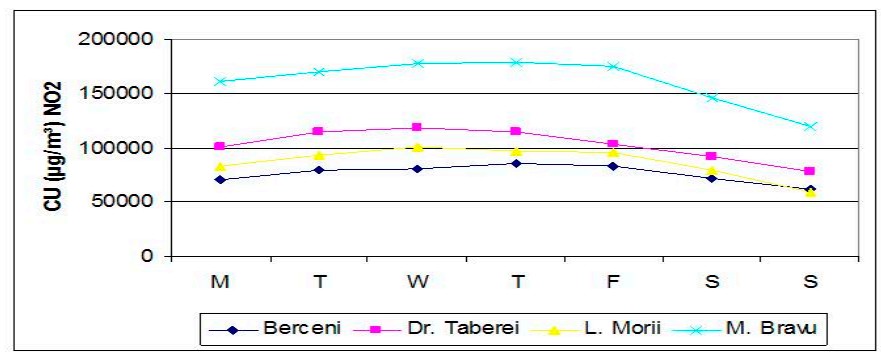

**Figure 13.** Evolution of accumulated quantities in different centers (CU) for NO$_2$ in days.

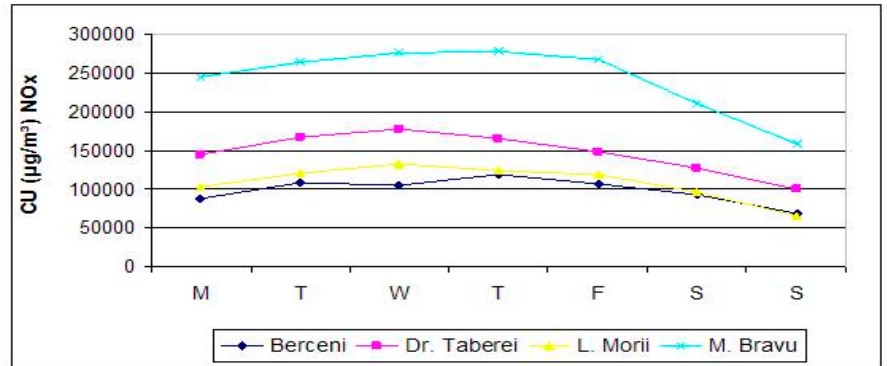

**Figure 14.** Evolution of accumulated quantities of NOx in different centers (CU) in days.

These predictions were necessary to establish the training program so that the start of the training schedule should begin in the days and hourly intervals with the lowest concentrations, after which the intensity and the allotted time for the exercise should increase gradually.

The values in the Figures 1–7, are cumulative for NO, NO$_2$, and NOx, and are expressed in μg/m$^3$, and the evolution of the concentration of pollutants in certain days and hours was monitored, due to the fact that the physical activities took place during a certain period of the day.

Additionally, it is necessary to emphasize the fact that at the age of 14–18 years, the students' bodies being in the period of transformation and maturation, the consumption of O$_2$ in the internal functional systems is increased, therefore the inhalation of a qualitative air, without pollutants, contributes to the acquisition a healthy lifestyle.

Figure 15 presents the mean annual concentrations of SO$_2$ resulted after pattern making. In the same manner of patterns (Figure 16), the map of iso-concentrations of SO$_2$ for the 3rd daily maximum value is presented (used as reference by EU directives of air quality).

Reporting to the imposed limits of EU directives requirements, it can be observed that the limit value is over-rated in the central city zone, both for the mean annual concentration for the ecosystem's protection (30 μg/m$^3$ comparing to VL = 20 μg/m$^3$), as for the 3rd daily maximum value.

The quality of the air in the environment has an influence on the time at which the equilibrium concentration of carboxyhemoglobin (COHb) is reached in the blood, which depends on both the carbon monoxide in the inspired air and on the volume of pulmonary ventilation. Thus, sustained physical activity depends on the initial value of carboxy-hemoglobin in the blood which, in the long run, contributes to maintaining the quality of life. The lack of physical activity and the inhalation of a polluted air can cause the occurrence of different nervous symptoms manifested through headaches, dizziness, digestive symptoms, and cardio-vascular symptoms, all starting with a COHb concentration of over 1% in the blood.

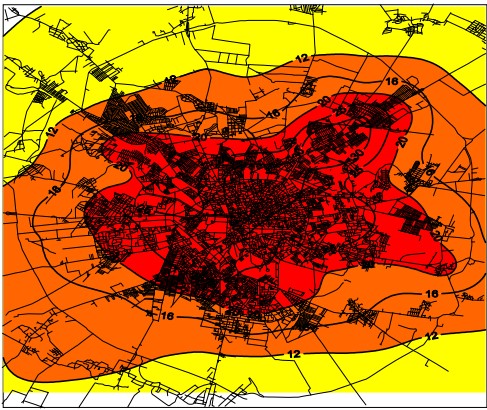

**Figure 15.** Mean annual concentrations of SO$_2$.

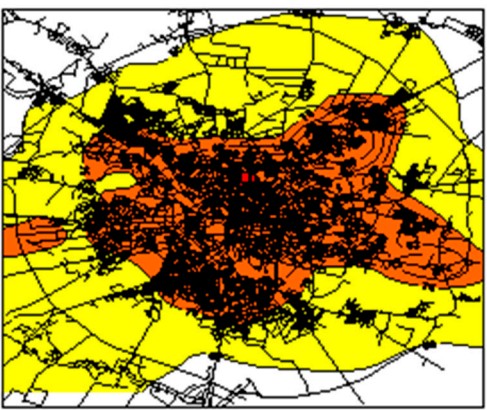

**Figure 16.** The 3rd daily maximum value of SO$_2$.

## 5. Discussions

The National Education System in Romania groups the students by age and distributes them by class, therefore, their biological development is relatively similar, and the students of the research study are in the value range 10.83–10.89 (9.84 + 0.99; 9.90 + 0.99), namely boys, and in the range value 11.40–11.51 (10.38 + 1.02; 10.49 + 1.02), namely girls. But from a social and informational point of view, there are major differences throughout years between high school students, and the low results obtained by HSS: Boys—59% (weak), 31% (satisfactory), and 10% (medium), girls—53.92% (weak), 34% (satisfactory), and 12.08% (medium), require finding solutions for this problem. The important role, in this aspect, is played by the physical education lesson, which develops skills in a pedagogical context, but the insufficiency of the number of hours allocated weekly tends to determine physical inactivity with personal/individual, but also social effects that lead to health deterioration [37].

"It is important that PETE educators work together with the physical education department of the local schools to support PETE students to effectively implement nonlinear informed approaches in a school environment" [38], just to try to correct that "34.88% of girls and 21.26% of boys of 14 years spent more time doing homework on school days and 24.32% of 14-year-old boys spent more time doing homework on days off" [39], which will affect the health status by acquiring diseases with repercussions on the quality of life [40–42]. Also, it is necessary to emphasize that between the age of 14 and 18 years is a period of transformation, and the consumption of O$_2$ in the internal functional systems of the organism is increased, therefore, the physical activities, as well as the inhalation of a qualitative air, without pollutants, contribute to the acquisition of a healthy life style. In this stage of development, generating ecological attitudes and sustainable actions have the role to contribute to the achievement of the objectives of the physical education lesson by acquiring new skills, education, and sustainable training for future generations.

The low values obtained by boys: 15 students (9th grade), 18 students (10th grade), 11 students (11th grade), 15 students (12th grade), representing 59%, and by girls: 10 students (9th grade), 17 students (10th grade), 14 students (11th grade), 16 students (12th grade), representing 53.92%, denote the level of physical training of the HSS assimilated in the physical education lessons. Thus, highlighting the correlation between the manifestation of the level of anaerobic effort capacity of the HSS and the air quality in the environment related to the concentration of pollutants in the atmosphere, in which the physical activities are carried out, determines the need to find alternatives to practice physical activities for maintaining a healthy lifestyle. The fact that, in Romania, it will not be possible to regulate, in the near future, the increase of the number of physical education lessons, these solutions are required to carry out sustainable physical activities in unpolluted air places [32,43]. By practicing any kind of physical activity, the resistance of the students' bodies increases, this aspect generating a conscious approach and a sustainable development in the information society.

According to European norms that support sustainable conditions for the performance of physical activities with a role in improving the quality of life, it can be observed that the limit value concentrations of $SO_2$ are over-rated in the central city area, both for the mean annual concentration for the ecosystem's protection (30 $\mu g/m^3$ comparing to VL = 20 $\mu g/m^3$), as for the maximum recorded value. Also, the values obtained underlined the cumulative evolution (CU) of the NO, $NO_2$, NOx pollutant concentration, recorded daily, in hours, in 4 centers, the closest to the high school. In this regard, the attention paid to climate change that influences the lifestyle, as well as the care taken to increase the quality of the breathable air by "placing the pollutant industrial parts far from the cities, as well as increasing the green surfaces influencing the meteorological phenomenon ensuring the rapid circulation of different gases ($CO_2$ consumption and releasing $O_2$)" [1] contribute to the promotion of physical activity in the open air, using natural environmental factors [44,45]. These aspects require a more aggressive reconsideration and promotion of the benefits of sustainable physical activities on HSS' health by guiding them to places/spaces where they can carry out extracurricular physical activities, in a sustainable way, in an unpolluted environment, and also by making them aware of the long-term impact on their lifestyle.

## 6. Conclusions

The low values registered by the students during the PEL allow us to conclude that the insufficient number of physical education lessons has a negative impact on the level of manifestation of the capacity of anaerobic effort, which impose the need to increase the degree of HSS' independence by carrying out extra-class physical activities.

Concentrations from the different compounds of the breathable air during a day in Bucharest showed a significant decrease in the time interval 13–19, which allows the scheduling of intense sustainable physical activities that lead to the increase of the life quality and to the improvement of daily activities. The more intense the physical activities in this type of interval are, the more the air inhaled by the HSS contributes to balancing the vital functions of the body with a role in their healthy development.

The applicability of this study will allow the reassessment of the level of manifesting the anaerobic effort capacity of the HSS from different schools, starting with the localities in which the authors of the present work activate and which will support the physical education teachers in the implementation of these investigations. In this way we will develop skills and we will broaden their horizon of knowledge and socialization in a sustainable way.

Taking into account the reduced number of physical education lessons provided in the Romanian school curriculum for high school students with effects on their formation, there is a need to do extra-class physical activities, which provide sustainability in maintaining lifestyle. Even if the number of physical education lessons is not be increased in the near future, teachers from faculties and universities will be involved and will collaborate with school principals through different partnerships and information programs to promote the major benefits obtained by practicing extra-class physical

activities at certain time when the concentration of polluting compounds in the breathable air is the lowest.

## 7. Key Limitations

Insufficient resources of information on how to efficiently record some values during the PEL, without affecting the structure of the lesson, have been negatively reflected in the rate of work we have initially intended in this study.

The difficulty of finding the number of students to be subjected to the study made the initial stage of forming the group to be investigated difficult. We have also encountered problems due to the reluctance of a large number of students to be part of the study because they felt that their exposure by performing these tests would affect their self-image compared to other colleagues.

Also, the instability of the students from the point of view of their personal beliefs, which are intensely manifested in this stage of their development, required an additional allocation of the authors' time for diversifying and motivating the speeches that supported the importance of doing the physical activities during the extracurricular period, since in the HSS' preferences, spending their extra-class time doing physical activities occupied the last place.

**Author Contributions:** Conceptualization, C.P.; formal analysis, D.-M.Ţ.; funding acquisition, B.F.; investigation, D.C.; Software, I.M.; Supervision, I.B.; writing—original draft, E.L.G. All authors have read and agreed to the published version of the manuscript.

**Funding:** This research received no external funding.

**Acknowledgments:** The high school students from "Mihai Viteazul" National College of Bucharest for data collection. This study was funded by the authors of paper who gratefully thank to the involved students.

**Conflicts of Interest:** The authors declare no conflict of interest.

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
