# Peer review of "Physical Education, Quality of Breathable Air and Their Effects on the Formation of High School Students as Sustainability in Maintaining the Lifestyle. Is the Physical Education Lesson Enough to Create Such Valences?"

_sustainability, doi:10.3390/su12031106_

Round 1

Reviewer 1 Report

Review

Title: Physical education and its effects on high school students as sustainability in maintaining the lifestyle. Is the physical education lesson enough to generate such valences

General comments

In introduction should made a many adjustments. Authors must better organize methodology, results and discussion. Please check the English. Many times I make an impression that some the words used are not correct.

Specific comments

Abstract

1 Page, line 25-29 Aim or aims? Rephrase first sentence, usually authors starts abstract from aims of the study. First sentence is too long and not clear

Second sentence the same, I don’t understand, divide this sentence

1 Page, line 34-37 Rephrase last sentence or add one more regards main conclusion… What is the main conclusion from this study?

Introduction

After I red this section, I was discouraged to view the rest of the paper. Of course, I looked at the whole article, but the introduction is too general, too long, most sentences are obvious. I did not find any justification, why authors prepared this article and I did not find what this paper add to the literature.

Try to shortcut this section, delete obvious sentences, try be more specific and add justification and sentences regards what in new, novel.

At the end should be aims of the study

Methodology

Page 4 line 196-197 only perform 3 vertical jumps? Is not too small? What is the brake between the jumps? After 3 jumps can increase muscle temperature? Participants were be optimal prepare after 3 jumps? Please add important information’s, because it is not clear.

At the end of this paragraph is citation no 36 – this is not scientific publication, I can not find, please change citation or add another

Graphics are part of results, move to section results

Please provide what statistical analysis were used by authors. How reviewer and future readers can assess value results from this study? Best way is to add another section Statistical analysis

Results and discussion

Results

If is possible please divide results and discussion section

In result section authors should provide only recorded values of different parameters.

Page 7 line 234-238 should move to the section statistical analysis

Page 7 line 249-252 this is not results but part of discussion, authors not present results but comment what can it result from, move to discussion

bio-psycho-social? Is this correct?

Table 1 what it is EEm, here you have two numbers, z test have one value? Maybe better to add legend under the table. If you observed any significant differences put *. Do this in every table and graphics if any significant differences exist

In section results authors should not cited another authors. Only present their own results

Discussion

Be more specific, compare your results with result another authors

In last paragraph highlight future directions in this field of study

Conclusion

Highlight practical application of this study in last paragraph

References - change style, incorrect style

Last comment – I think the title should be little changed, authors pays a lot of attention to air pollution, maybe this aspect should be added to the title

Author Response

Thank you!

Reviewer 2 Report

This study was to highlight the correlation between the level manifestation of physical condition by anaerobic effort capacity for the HSS and the quality of the air from the environment in which the physical activities take place, by guiding them to practice in places and spaces that offer sustainable conditions with implications for increasing the quality of life and at the same time, knowing their evolution of this capacity of effort, to determine an optimal selection for physical exercises to maintain the lifestyle for HSS and to guide them to practice different sports at the performance level, all harmonized in a friendly way with environment. Also, authors want to present the direct connection between the number of hours of physical education and sport class of school program and the value that these lessons have on a harmonious and psycho-physical development of high school students and the importance of breathable air quality during the development of physical activities and acknowledging the benefits of sustainable environment in which they daily activities. I consider that there are some specific changes and suggestions that should be made to improve the quality of the paper.

GENERAL SUGGESTIONS

- The manuscript abstract is too weak. The authors do not report the methods, the statistical analysis, the main results and the main conclusions, please rectify.

-   The manuscript does not show the hypotheses of study, please rectify.

-  Authors should correct throughout the manuscript all abbreviations. The first time it appears in the manuscript is in full, with the abbreviation in parentheses and only after that only the abbreviation is used.

-    Insert the anthropometric characteristics of the subjects, eg weight, height and age.

-    What were the specific PEL exercises that the authors used for warm-up? Insert in manuscript

-    Did the authors have any familiarization with a sample of the tests that would be performed in the study? If yes, put in the manuscript. If not, why not do it?

- Insert a section for statistical analysis only and describe the procedures performed.

- The first paragraph of the discussion should be the purpose of the study. Rectify.

- Insert a section with key limitations of the manuscript.

- References are not formatted according to journal guidelines, please rectify.

- The manuscript is a little confusing, as the authors put some information that should be inserted in another section, eg put in the results information that should be in the section of the statistical analysis.

Author Response

Thank you!

Reviewer 3 Report

I found the depth of this research design and results very significant. The number of subjects were within recommended suggested quantitative research designs.The topic a major concern today to support the effects of Global Warming on child development and the future for growing up to continue physical activity but in a healthy environment. The conclusion had ideal recommendations for participants to take with them from being in the study to look to future application to best conditions for physical activity in the environment. I would suggest in the conclusion how can future educators take this information and apply to their environments for their students. The  conclusion did state application for future healthy lifestyles knowing this information about quality of air and other factors in the environment 

Author Response

Thank you!

Round 2

Reviewer 1 Report

Review  after corrections

Title: Physical education, quality of inspired air and their effects on the formation of high school students as sustainability in maintaining the lifestyle. Is the physical education lesson enough to create such valences?

General comments

Now article is more clarity, but still have some problems. Please make new adjustment.

Specific comments

Title: my reflection raises a word “quality of inspired air” especially “inspired” – I don’t understand

Abstract: first sentence ??? … highlight – maybe better investigate. Maybe better to divide to two aims. I not feel connection between title and first sentence in abstract

Line 78 – first sentence of the paragraph is bolded – please delete

Now introduction is much better, possible is to  follow what the authors wanted to provide.

Still have a problem with connection with aims, hypothesis and title, I suggest to modified a title and small adjustments in aims and hypothesis.

Methodology

Better to understand, warm-up better to understand

172-177 Author do not need to repeat all the time “with return to the starting place by walking-active pause” – write only in the first sentence

Statistical analysis

Big mistake in this section - anthropometric characteristic of the participants please move up to methodology section, because it is not statistical analysis. Lines 209-230 is ok

Results

This section is much better. For me is too many figures and little too long, to many results, but if authors can not deleted some part of this section it is ok

Discussion

Authors provide a many results, the discussion is too shallow. Please discuss more results

Conclusions

Too long. Divide the section and create practical application or some of conclusion move up to discussion and expand

Author Response

Thank you!

Round 3

Reviewer 1 Report

The corrections improved the quality of the text. The discussion is too short and shallow. Conclusions are fine

Author Response

Thank you!
